# Towards a Fully Automated Scanning Probe Microscope for Biomedical Applications

**DOI:** 10.3390/s21093027

**Published:** 2021-04-26

**Authors:** Witold K. Szeremeta, Robert L. Harniman, Charlotte R. Bermingham, Massimo Antognozzi

**Affiliations:** 1School of Physics, University of Bristol, Tyndall Avenue, Bristol BS8 1TL, UK; w.szeremeta@bristol.ac.uk (W.K.S.); cbermingham@hotmail.co.uk (C.R.B.); 2School of Chemistry, University of Bristol, Cantock’s Close, Bristol BS8 1TS, UK; rob.harniman@bristol.ac.uk

**Keywords:** SPM, automation, femtonewton resolution, vertical probes, translation stages, inertial drive, piezo actuators, vertical positioning

## Abstract

The increase in capabilities of Scanning Probe Microscopy (SPM) has resulted in a parallel increase in complexity that limits the use of this technique outside of specialised research laboratories. SPM automation could substantially expand its application domain, improve reproducibility and increase throughput. Here, we present a bottom-up design in which the combination of positioning stages, orientation, and detection of the probe produces an SPM design compatible with full automation. The resulting probe microscope achieves sub-femtonewton force sensitivity whilst preserving low mechanical drift (2.0±0.2 nm/min in-plane and 1.0±0.1 nm/min vertically). The additional integration of total internal reflection microscopy, and the straightforward operations in liquid, make this instrument configuration particularly attractive to future biomedical applications.

## 1. Introduction

Over the past few decades, the discovery of new nanoscale phenomena has produced scientific and technological breakthroughs across various disciplines from natural sciences to engineering [1] and medicine [2,3,4]. The increased interdisciplinarity of these discoveries, and their comparable length scale, reveals a convergence at the nanoscale between these different disciplines [5,6] with the promise for even stronger future integration. In medical science, the transition from microscale to nanoscale observations has enabled faster, more accurate and cheaper diagnostic tools [7]. Whether it is the relation between cell wall mechanics and cancer cells [8,9,10,11,12,13,14] or the detection of cellular nanoscale fluctuations [15,16,17], it seems clear that the exploitation of nanoscale effects is crucial to the future of biomedical diagnostics.

Scanning Probe Microscopy (SPM), one of the most diverse techniques in this area, has become an essential research tool in cellular and molecular biology [18]. When operating in force mode, Atomic Force Microscopy (AFM) [19] extends the intuitive sense of touch directly from the user’s frame of reference down to the nanoscale. As a result, AFM data add a unique mechanical assessment of the specimen that is lacking in other characterisation techniques, such as Dynamic Light Scattering [20] (DLS), Raman [21], or Nuclear Magnetic Resonance [22,23] (NMR) spectroscopy. It is precisely the combination of extreme spatial resolution and the ability to manipulate the sample at the nanoscale, that gives AFM the potential to become an ideal tool for biomedical applications [24,25,26]. Despite these capabilities, it has been challenging to transition SPM technology from research laboratories to clinical or medical applications [27]. The main difficulties can be summarised in three points: (1) the technology requires a highly skilled specialised operator to be present at all times, (2) the most advanced SPM applications are difficult to reproduce [14], and (3) the statistical basis underpinning SPM measurements can be considered weak from a biomedical perspective due to the low throughput [28].

Here, we suggest that a fully automated SPM with femtonewton sensitivity and the ability to perform measurements in physiological conditions could solve most of the above problems and enable widespread adoption of this technology in the biomedical area. It is essential to notice that the requirements for automation and a bio-compatible environment have to preserve the unique SPM sensitivity and versatility to ensure the full impact of this technology.

This work presents a bottom-up design approach to the challenges described above and its initial implementation on an actual Lateral Molecular Force Microscope (LMFM) [29]. The LMFM uses vertically oriented probes (VOP) with a much higher force sensitivity than conventional horizontal AFM cantilevers due to their orientation [30]. The sensitivity of the LMFM has already been successfully demonstrated [31] and is essential when investigating weak biomolecular processes [32,33] or imaging soft nanostructures [34]. Presented below are a series of design solutions, which significantly increase the stability, precision and usability of the modified LMFM. Finally, these changes align with the constraints imposed by future automation levels and retain the versatility required for biomedical applications.

## 2. Materials and Methods

### 2.1. Sensitivity and Versatility of Vertically Oriented Probes

LMFM [29] is part of the SPM family with a vertically oriented probe and can use commercially available ultra-compliant silicon nitride cantilevers (NuNano, Ltd, UK) [35]. Standard AFM probes are mounted horizontally and are at risk of jumping into contact with the surface due to attractive interaction forces between the tip and the substrate [36]. This effect limits the minimum cantilever stiffness necessary to counteract the force gradient above the surface [37]. A vertically oriented cantilever benefits from the fact that the minimum stiffness needed to prevent a jump-to-contact has an angular dependence given by $cos^{2}\left(90^{\circ }-\theta \right)$ [38], where θ is the angle of the cantilever from the vertical, as shown in Figure 1. A vertical orientation enables more compliant cantilevers resulting in greater in-plane force sensitivity and a much greater tip-sample separation control. The silicon nitride probes routinely used in this work have a minimum stiffness of the order of $10^{-6}$ N/m–$10^{-5}$ N/m, compared with $10^{-3}$ N/m–10 N/m for a standard AFM cantilever. LMFM cantilever stiffness is comparable to optical tweezers stiffness with the advantage of a vertical positional control with sub-nanometre resolution. Additionally, LMFM maintains the ability to image the surface of very soft samples in a liquid environment [34].

### 2.2. Scattered Evanescent Wave Detection System

The small size of LMFM cantilevers [35], and their vertical orientation, makes it difficult to use conventional AFM optical detection systems [18,19], so LMFM detects the probe’s tip position using the Scattered Evanescent Wave (SEW) detection system [29]. The SEW detection is based on the fact that objects entering the evanescent field scatter the light, transforming it from near-field to far-field. The scatterer’s three-dimensional position is subsequently measured using a four-sector photodetector. The SEW system has been successfully tested in ambient and liquid environments [35]. The evanescent wave is created upon total internal reflection of a laser beam at the glass-air (or glass-water) interface. The LMFM setup described in Figure 1 makes use of two fibre-coupled lasers; laser #1 is a OBIS FP 660LX (Coherent, Inc., Santa Clara, CA, USA) and laser #2 is a Stradus VersaLase (Vortran Laser Technology, Roseville, CA, USA) with multiple wavelengths (488 nm, 561 nm, 642 nm). The two perpendicular laser beams produce two concentric evanescent fields with perpendicular wave vectors. The SEW detection system uses one wavelength (642 nm-laser #2), while the other laser lines can be used to study optical forces in evanescent fields [31] or combine SPM with Total Internal Reflection Fluorescence (TIRF) microscopy. Two adjustable mirrors are located underneath the TIRF objective lens (Nikon, 100x TIRF objective NA=1.49) and direct the laser beams into the lens. Changing their position affects how far from the axis of the objective lens the beam propagates, which, in turn, affects the angle (α) of incidence of the beam onto the glass surface. Two further mirrors are positioned underneath the objective to stop the exit beams from reaching the detectors. When the tip of the probe scatters the evanescent field, the TIRF objective lens collects the light and produces a high-magnification image in the detection plane. Before reaching the four-sectors photodiode (Hamamatsu S5990-01), the scattered light goes through a dichroic mirror which separates the detection wavelength from the other wavelengths. A second objective lens (Olympus, Plan Fluorite Oil Immersion Objective NA=1.30) is positioned directly in front of the photodetector for further angular magnification. An sCMOS camera (Hamamatsu Orca Flash 4.0) is used for conventional evanescence light scattering microscopy or TIRF microscopy. An automatic routine uses the vertical probe image from the camera with the signal from the photodiode to position the cantilever on the optical axis of the TIRF lens. This operation uses the motorised stages described in the following section.

### 2.3. Design of Position Control Stages Compatible with Automation

Remote control of sample and probe position is the initial step towards autonomous operations, and it demands reliable, precise, and stable positioning mechanisms. Furthermore, designing a fully autonomous SPM system requires consideration of how the positioning system interfaces with an automatic sample and tip exchange.

After the invention of the first dynamic piezoelectric translation stage by D.W. Pohl in 1986 [39], the stick-slip design has become one of the most commonly used techniques in SPM for remote operations [40,41,42].

The upgrade of the LMFM described here incorporates seven motorised degrees of freedom (DoF). The probe Vertical Positioning System (VPS) uses a new shuttle-and-tube design with stick-slip actuation (Figure 2). The Horizontal Positioning System (HPS) for the sample and the probe uses a stick-slip translation stage (Figure 3) inspired by Drevniok et al. [43]. The sample and the probe positioning systems are both connected to the microscope focusing plate. This plate moves in the vertical direction to adjust the microscope’s focus. Three DC-motors, with magnetic encoders, are embedded in the microscope’s base and provide the focusing movement.

To produce stick-slip motion in the vertical direction, the acceleration a(t) of the actuator (i.e., piezoelectric crystal) needs to satisfy the following relation:(1)$$\left|a\left(t\right)\right|>|\frac{F_{s}}{m}+g|,$$
where *g* is the acceleration due to gravity, $F_{s}$ is the static friction, and *m* is the mass of the moving part. For horizontal movements, we can disregard the acceleration due to gravity.

The VPS design consists of 3 main elements: (1) a moving shuttle (a stainless steel cylindrical plug, 4 mm in diameter and 7 mm long), (2) a smooth glass tube (an NMR glass tube with 4 mm inner diameter and a length of 14 mm), and (3) a piezoelectric stack actuator (PK4FA2H3P2 from Thorlabs, Inc., Newton, NJ, USA). Figure 2a shows a diagram of the VPS. A micro-fabricated cantilever is mounted on the stainless steel plug, which fits into the glass tube. A small amount of Teflon tape is wrapped around the plug to adjust the static friction between the tube and the plug. The glass tube is directly connected (i.e., glued) to the piezoelectric stack actuator and acts as a guide rail for the cylindrical shuttle. The static friction between the shuttle and the glass tube is sufficient to counteract the gravitational pull and keeps the shuttle stationary. At the same time, the friction coefficient is sufficiently low to allow for the slip to happen when the actuator’s acceleration satisfies Equation (Equation 1). In other words, a sufficiently rapid movement of the glass cylinder in the upwards (downwards) direction produces a slippage of the plug downwards (upwards) moving the plug towards the bottom (top) end of the tube. When an asymmetric sawtooth waveform is applied to the actuator, the plug starts moving [44]. A modified sawtooth waveform, with exponential deceleration, (Figure 2c) produced the most consistent results in terms of step size and plug speed. The continuous (non-stick-slip) vertical extension of the piezoelectric actuator is used for fine vertical positioning of the probe (with 0.1 nm resolution).

Figure 2b describes some of the modes in which the VPS can work. The VPS can operate in a liquid medium and the tube around the probe acts as a container for the liquid (steps 1–4). Different cylindrical probes can be mounted to perform Near-field Scanning Optical Microscopy (NSOM) or Scanning Tunnelling Microscopy (STM) (5 and 6). The VPS can operate reliably in a vacuum of $10^{-6}$ mbar (7). The VPS can be used to deposit (regain) the plug to (from) a second cylinder. This capability can lead to an automatic probe exchange as described in the discussion section (8–11).

The lateral positioning of the probe and sample is achieved using two concentric Horizontal Positioning Systems (HPSs), as shown in Figure 3a. The inner HPS supports the sample, while the outer stage houses a tilting bridge carrying at its centre the Vertical Positioning System for the probe (Figure 3c,f).

Each HPS comprises two stacked plates, a bottom B-plate and a top T-plate and sits on three shear-piezoelectric motors (PN5FC2 from Thorlabs, Inc., Newton, NJ, USA) with ball bearing ends (Figure 3b). The three piezoelectric actuators can move in the *x* and *y* directions (Figure 3d) and they couple with the B-plate of the HPS via a 3-point kinematic mounting. The B-plate lower side contains a sapphire disk and two aligned pairs of parallel cylinders that contacts the three ball bearings and constrain the plate movement along the *x*-axis. The B-plate top side has the same configuration as the lower side but rotated by $90^{\circ }$ (Figure 3e). Three ball bearings on the lower side of the T-plate couple with the B-plate constraining the T-plate movement in the *y*-axis. All six piezo actuators (three for each HPS) are fixed on the microscope’s focusing plate (component 2 in Figure 3d) in the same orientation, ensuring a mechanical link between the sample and probe HPSs. Consequently, focus adjustments do not change the tip-sample separation—a critical requirement when the probe is a few tens of nanometres away from the sample.

A similar sawtooth waveform to the one shown in Figure 2c is used in the HPS to produce the stepping action for long-range movements (millimetre range), while the conventional piezoelectric extension (7μm range) is used for fine positioning and scanning (with 0.1 nm resolution). The sample’s position is measured with 1 μm resolution using two optical encoders (Atom${}^{TM}$ from Renishaw plc, UK). The system shown in Figure 3f has a footprint of 400 cm${}^{2}$ and a range of ±1 cm for the sample stage. It is essential to notice that the top plate in both HPSs is kept in position only by gravity allowing it to be easily lifted and replaced.

Due to its compact size, the VPS can be easily integrated into a bridge design with motorised tilt, as can be seen in Figure 4a. Two opposing stepper motors are fixed on the top plate of the probe’s HPS via the central shafts. When powered, the outer casing of the motors rotates rather than the shafts. The stepper motors are integrated into the two piers of the bridge, causing the bridge to tilt with an angular resolution of $0.17^{{}^{\circ}}$. The axis of rotation, determined by the shafts’ position, is at the level of the sample surface. In this way, when the cantilever’s tip is close to the sample, tilting the bridge does not produce any lateral movement of the tip. In other words, the cantilever pivots around its tip.

To obtain consistent results in LMFM, it is essential to determine the cantilever’s vertical tilt within a fraction of a degree. An acoustic actuator is used to oscillate the cantilever out of resonance as this ensures stable phase measurements. When the cantilever is not vertical, this oscillation causes the tip to move laterally and up and down with respect to the surface. The vertical oscillation can be observed as a modulation in the sum signal detected using the SEW method (see Figure 4b,c). A lock-in amplifier can measure this oscillation amplitude and its relative phase. A large cantilever tilt angle corresponds to a large oscillation of the sum signal. The cantilever tilt is adjusted by minimising the amplitude measured by the lock-in amplifier. As illustrated in Figure 4b, when the probe is tilted in one direction, the sum signal has a specific phase relation with the acoustic signal. If the tilt direction changes, the phase shifts by approximately $180^{\circ }$. The vertical position of the cantilever can be estimated within $0.17^{\circ }$ using an automatic routine that finds the minimum amplitude with an associated sudden change in the phase signal (see Figure 4c).

### 2.4. Force Measurements in Intermittent Mode

The LMFM low-compliant cantilevers can be used to measure extremely small in-plane forces. In this mode, the applied force is periodically switched on and off, while the correspondent distribution of the cantilever’s positions is recorded. The force is then calculated by multiplying the difference between the two distributions’ means by the cantilever’s spring constant. The precision of these measurements can be increased by accumulating several on/off cycles, while the accuracy is determined by the error in the spring constant *k*. The spring constant is determined experimentally from the power spectrum density (PSD) of the thermally fluctuating probe (not actively driven) which can be described by a Lorentzian function [45]
(2)$$S_{x}\left(f\right)=\frac{k_{B}T}{\gamma \pi^{2}\left(f_{c}^{2}+f^{2}\right)’}$$
where $k_{B}$ is Boltzmann’s constant, *T* is the temperature of the surrounding environment, $f_{c}$ is the corner frequency, and γ is the drag coefficient. The cantilever’s spring constant can be found by fitting the experimental PSD with Equation (Equation 2) and knowing that $k=2f_{c}\pi \gamma$. By repeating this process multiple times, one can obtain a distribution of *k* values, the mean of which is used to calculate the force experienced by the probe.

Optical forces caused by an evanescent field [31] were measured using ultra-compliant silicon nitride cantilevers with spring constants in the range between $10^{-6}$ N/m and $10^{-5}$ N/m. The position of the tip was recorded at a sampling rate of 48 kHz. The data were subsequently decimated by a factor of 6 to account for an auto-correlation time of 0.125 ms [46,47,48]. Laser #2 (561 nm) in Figure 1 was switched on and off using a TTL signal with a frequency of 1 Hz.

## 3. Results and Discussion

### 3.1. Microscope Positioning Resolution

The VPS combines a stick-slip mechanism (coarse positioning) and a continuous voltage actuation (fine positioning). In stick-slip mode, it is possible to vary the step size allowing for millimetre-long movements and nanoscale positioning. For finer vertical positioning (e.g., less than 10 nm), the continuous voltage mode is used. Figure 5a shows the smallest steps obtained using the VPS when positioning the probe in the vertical direction. The smallest steps towards the surface (’steps down’) were recorded to be $S_{d}=2.79\pm 0.09$ nm with a standard deviation of $\sigma_{d}=0.52$ nm. The smallest steps away from the surface (’steps up’) had a value $S_{d}=2.7\pm 0.1$ nm with a standard deviation of $\sigma_{d}=0.7$ nm. Similar values are obtained when the system operates in water. The smallest step size is almost two orders of magnitude smaller than the decay length of the SEW detection system’s evanescent field, ensuring a safe approach of the tip to the surface using the stick-slip mode exclusively.

### 3.2. Microscope Stability

The microscope’s overall stability can significantly improve performance for measurements with low signal-to-noise ratio, and where long integration time is required. The use of kinematic mounts and the symmetric arrangement of the constraints result in drift values comparable to commercial AFM systems [49]. The measured mechanical drift of the probe and sample positions (see Figure 5b) had an in-plane velocity of 2.0±0.2 nm/min for both the probe and the sample. A smaller drift velocity of 1.0±0.1 nm/min was recorded in the vertical direction for the VPS.

The piezo actuators position in the microscope focusing plate further increases the overall stability of the HPS. The sample top plate does not have any electrical connection and relies on gravity to connect to the kinematic mount. This design allows for easy sample exchange that can potentially be automated. A conveyor belt solution, similar to the ones used for silicon wafer handling, could be implemented to automate this process (see Appendix A).

Using the sample stage as an SPM scanning unit, the cross-talk between the fast and slow scan direction was minimised by applying a linear correction in the orthogonal axis. Figure 5c shows that the cross-talk between the two scan-axes after compensation is approximately 0.3%.

### 3.3. Microscope Force Resolution

Combining cantilever’s stiffness between $10^{-6}$ N/m and $10^{-5}$ N/m with sub-nanometre sensitivity of the probe’s position, enables force measurement with sub-femtonewton resolution. Figure 6 shows the Gaussian distributions of positions of the cantilever with and without an applied optical force. The measurements are obtained by illuminating the cantilever tip for 16 min with an intermittent evanescent field at a rate of 1 Hz. The shift between the two distributions is equal to 0.52 nm ± 0.03 nm that corresponds to an optical force of 2.7 fN ± 0.2 fN.

A time accumulation longer than 1 min results consistently in a sub-femtonewton standard error. On the opposite side of the spectrum, the system’s stability allowed a 3-hour-long measurement of an evanescent optical force of 0.3 fN ± 0.1 fN. The force resolution at these accumulation times reaches a limit due to laser intensity fluctuations.

So far, we have shown how an original SPM bottom-up design can combine remote control of the instrument with extreme force resolution and stability. This approach preserves various SPM modes (e.g., STM and NSOM) and can use commercially available, ultra-compliant, micro-cantilevers. As with any design solution, this particular solution has intrinsic constraints; the SEW detection method requires a transparent surface, whilst the decay length of the evanescent field confines the tip’s position to within 150 nanometres from the optical interface. On the other hand, many biomedical applications may benefit from a transparent substrate and integration with fluorescence microscopy. Furthermore, simple and effective operations in a liquid environment may have priority over other aspects.

Usually, AFM operating in a liquid environment is deemed unsuitable for the femtonewton and sub-femtonewton force regime [50]. This limitation is only valid when using standard micro-cantilevers in a horizontal orientation. Several examples in the literature have demonstrated that ultra-compliant custom-made cantilevers in a vertical orientation can operate in a much lower force regime [51,52,53,54,55] and almost any environment [32,56]. For the use of this technology in the biomedical area and based on the cantilevers’ unique properties, we envisage three main types of operation. Firstly, the imaging of soft nanostructures or biomolecules in a liquid environment using a constant-height dynamic mode. This method has shown clear advantages over conventional AFM in observing self-assembling peptide cages [34] and provides molecular resolution with simpler operations [57]. Secondly, the in-situ detection of conformation changes in cell membrane proteins triggered by ligand binding to a specific receptor domain. This approach has already given a unique insight into the binding mechanism of Moraxella catarrhalis, unveiling adhesin UspA1 conformational change upon binding to fibronectin and CEACAM1 [33]. Thirdly, an in-vitro force spectroscopy mode in which conformational changes in proteins tethered between the cantilever and the surface are directly observed. The sub-nanometre control of the tip-sample separation, when using vertically oriented cantilevers, provides a unique advantage amplified by the possibility of using the large set of AFM force spectroscopy tools. This mode of operation has successfully detected the stepping action of single kinesin molecules processing on microtubules [32]. In this experiment, motor proteins are first bound to the cantilever and then moved closer to an immobilised microtubule. As soon as one protein starts processing, the microscope records the changes in the cantilever position, revealing the individual steps produced by the kinesin protein. Recently, this technique has been updated using small cylindrical glass cells around the cantilever that facilitate the cantilever incubation with kinesin molecules and its handling [58]. The VPS described here builds on this improvement and radically simplifies this type of experiments by providing an automatic approach routine with integrated vertical angle correction. The HPS with optical encoder further simplifies the positioning of the cantilever directly above immobilised microtubules.

It becomes apparent how the bottom-up design here presented builds on the already demonstrated capabilities of LMFM introducing significant advantages to the various experimental protocols, while simultaneously improving the force sensitivity and stability of the overall instrument. The following section will show how automation and machine learning are interweaved into this bottom-up design, enabling a new generation of autonomous SPMs with an ultralow-force regime.

### 3.4. Implementing Full-Automation and Artificial Intelligence

Automation of the SPM experiments is key to reduce the operator’s bias, improve reproducibility of results and enable widespread adoption of this technology in the biomedical area [59].

One can divide the SPM workflow into different levels of automation. The first level is related to sample preparation and cantilever functionalisation for biomedical applications. Here, the technology supporting robotic-assisted assays used in high-throughput screening can provide the necessary solutions to reach this level of automation. The second level of automation includes sample and cantilever exchange. This level is essential to enable 24/7 continuous operations, and future integration with artificial intelligence (AI) [60]. Incidentally, this level is the most challenging in terms of instrument design. Most, if not all, the commercial SPMs available today are not compatible with this type of automation. The best examples of SPM with this level of automation are currently tailored for vacuum environment or semiconductor industry [61], but they would be unsuitable for biological samples.

Figure 7 describes a possible workflow in which the specific characteristics of the VPS and HPS presented here are used to ensure the first two levels of automation. More specifically, The HPS can use a conveyor belt system to enable sample exchange. First, the HPS is temporarily lowered, connecting the top (sample) plate with the conveyor belt that moves it away from the microscope. Then, a new sample plate can be loaded, performing the same actions in reverse (see Appendix A). For automated probe exchange, the unique VPS ability of expelling and loading a shuttle carrying a probe means that we can use two VPSs to split the functionalisation step from the measurement step. A first VPS (i.e., functionalisation VPS) can be used to pick up a new plug with cantilever, perform various functionalisation steps and deliver the plug on a carrier plate. The plate is then transported to the microscope HPS as if it was a new sample. At this point, the microscope VPS can load the new plug and the carrier plate can be removed from the microscope (see Appendix A).

The fully motorised microscope, as described in the text, allows the implementation of the third level of automation through AI-driven operations. Recently, significant progress has been made in the automation of the AFM data collection process [59], including integration with AI-based solutions [60,62]. The significant similarities in imaging mode between conventional non-contact AFM and LMFM [34,57] imply that the AI-based solutions for AFM could be adapted for LMFM image acquisition modes. We predict that a fully motorised SPM will benefit considerably from rapid advances in AI-assisted data collection. The last level of automation is related to SPM data analysis. Application of AI to this level is already giving promising results in biomedical applications [14,63]. Rapid progress should be expected in this area, considering the vast and growing library of machine learning algorithms dedicated to interpreting images.

In conclusion, we have shown how an implementation of a bottom-up design for a particular type of SPM can combine femtonewton sensitivity with automation and become the first step towards a new class of instruments that operate fully autonomously in the force regime and environment ideal for biomedical use. Moreover, leveraging high throughput in SPM data collection and analysis will enable progressively more effective AI algorithms to transform medical research and diagnostics.

## Figures and Tables

**Figure 1 sensors-21-03027-f001:**
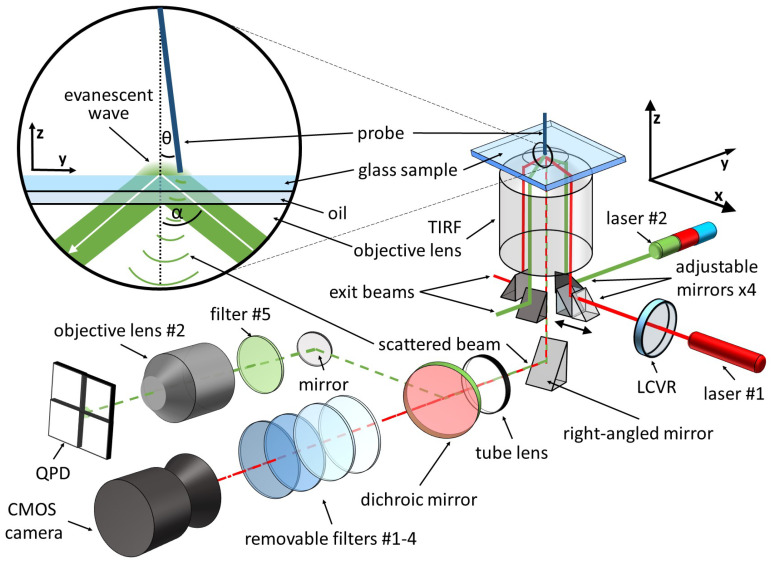
Diagram of the LMFM set up, including the SEW detection system. A closeup diagram shows the vertically mounted cantilever forming an angle (θ) with the normal. A totally internally reflected laser beam, forming an angle (α) with the normal, creates an evanescent wave on the sample side.

**Figure 2 sensors-21-03027-f002:**
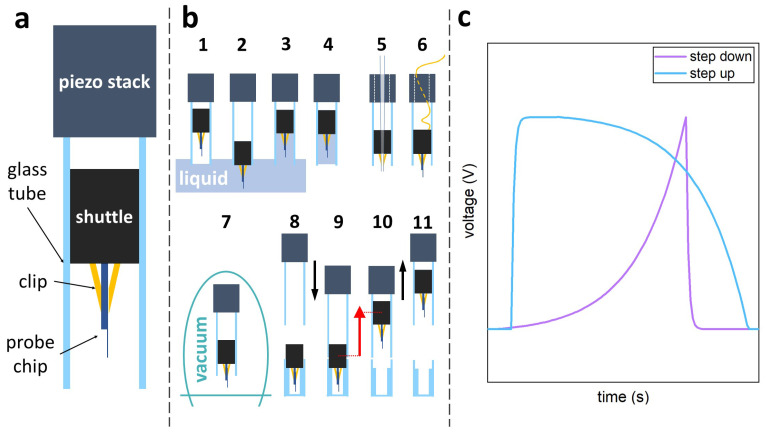
(**a**) Schematic drawing of the Vertical Positioning System (side view) (**b**) schematic drawings of different applications of the VPS. (1–4) The VPS has been tested in liquid environment, and the glass tube can act as a functionalisation chamber. (5) The VPS can be used for other SPM techniques, e.g., NSOM, which uses tapered waveguide probes or (6) STM by establishing an electrical connection to the conductive probe. (7) The VPS has been tested and is compatible with working in a vacuum of $10^{-6}$ mbar. (8–11) The VPS can expel and regain the plug under simple stick-slip operations. This simple probe exchange can be automated. (**c**) Asymmetric sawtooth waveforms with exponential deceleration used to drive the VPS.

**Figure 3 sensors-21-03027-f003:**
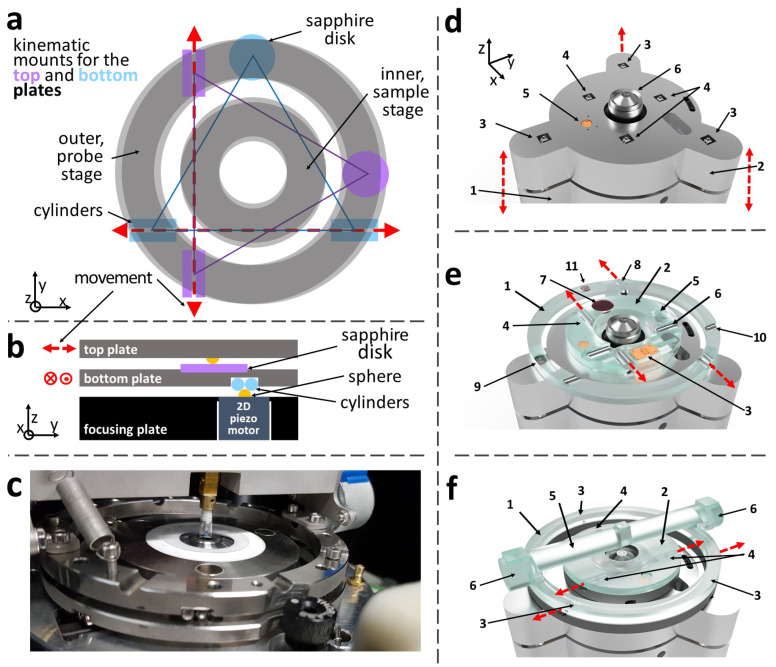
(**a**,**b**) Top view of the probe and sample HPS and kinematic mount details. Only the outer ring is described. Two sets of three ball bearings are positioned at the vertices of two orthogonal equilateral triangles. The first set is fixed on the piezo motors, while the second set is mounted on the top plate’s lower part. The bottom plate can only move in the *x*-direction under stick-slip action, while the top plate’s movement is confined to the *y*-axis. (**c**) Photograph of the current implementation of the HPS and the VPS. (**d**–**f**) CAD drawings, including the microscope’s focusing plate and the sample and probe HPSs. (**d**) (1) Microscope’s base where the objective lens (6) is mounted. (2) Focusing plate which can move in *z*-direction using three DC-motors (not shown). The two sets of three XY piezo motors, (3) and (4), are mounted on the focusing plate (2). (5) Otical encoder that measures the movement in the *x*-direction of the sample stage. (**e**) HPSs’ bottom plates for the probe (1) and sample (2). The two pairs of lower cylinders (8) and sapphire disk (9) ensure ring (1) can only move in the *x*-direction. Similarly, ring (2) can only move in the *x*-direction thanks to the pairs of lower cylinders (4) and sapphire disk (5). The same type of kinematic mount is present on the two rings’ top surface to ensure both top plates can only move in the *y*-direction (cylinders (6) and (10) and sapphire disks (9) and (7), respectively). The optical encoder (3) measures the movement of the top sample plate in the *y*-direction. (**f**) Top probe’s (1) and sample’s plate (2). The mechanical coupling between the top and bottom plates is ensured by two sets of three ball bearings (3) and (4) mounted on the lower surface of the probe and sample plate, respectively. The Vertical Positioning System is partially visible at the centre of the bridge (5). The tilt of the VPS is controlled by rotating the two stepper motors (6).

**Figure 4 sensors-21-03027-f004:**
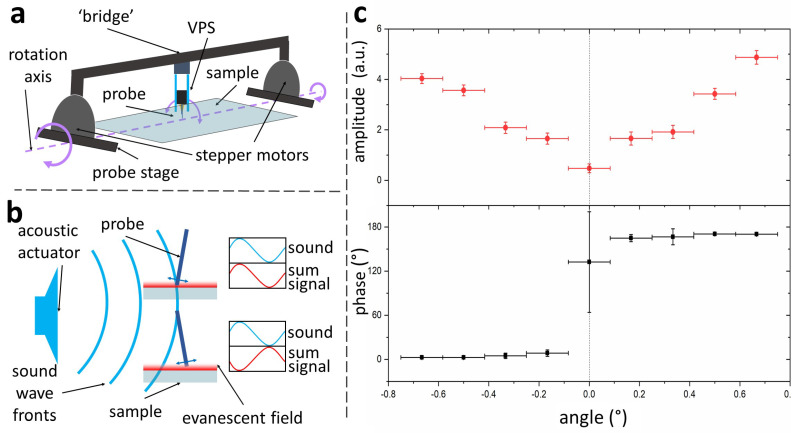
(**a**) Diagram of the probe’s angular adjustment method. Two opposing stepper motors are embedded in the bridge carrying the VPS. A rotation of the motors causes a tilt in the bridge changing the vertical orientation of the cantilever. The tilt axis (dashed line) is level with the sample surface, so the cantilever pivots around its tip when changing the tilt. (**b**) Acoustic actuation of the probe can be used to determine its exact vertical position as described in the text. (**c**) Experimental results showing how the amplitude of vertical oscillation and relative phase change as a function of the probe’s angle. Upon going through the vertical position (tilt angle $\theta =0^{{}^{\circ}}$), a sharp change in phase is observed. This position also corresponds to the minimum amplitude of vertical oscillations.

**Figure 5 sensors-21-03027-f005:**
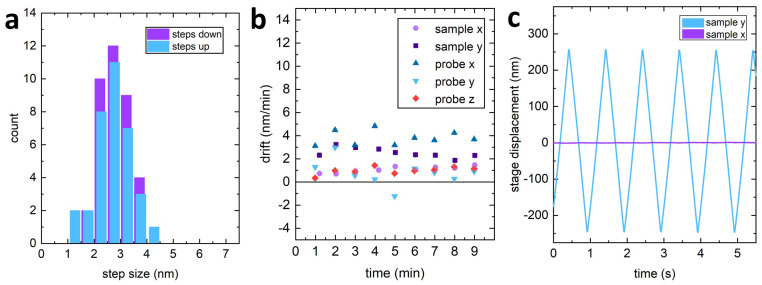
(**a**) Histogram of the average step size of the VPS in both upwards and downwards directions. (**b**) Measurements of the sample and probe’s drift in the xy-plane and of the probe’s drift in the vertical direction. (**c**) The cross-talk between displacements in orthogonal directions of the sample’s HPS is around 0.3%.

**Figure 6 sensors-21-03027-f006:**
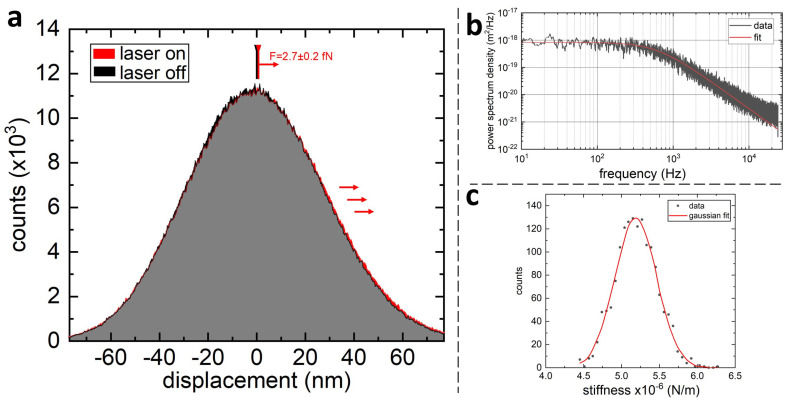
Force measurement with sub-fN precision. (**a**) A shift in the positive direction of the position distribution upon laser illumination (red) is visible to the right of the grey area which corresponds to the overlap of the two distributions. (**b**) A PSD of the probe’s position used to evaluate the probe’s stiffness. (**c**) A distribution of stiffness values for the cantilever used to calculate the force in (**a**).

**Figure 7 sensors-21-03027-f007:**
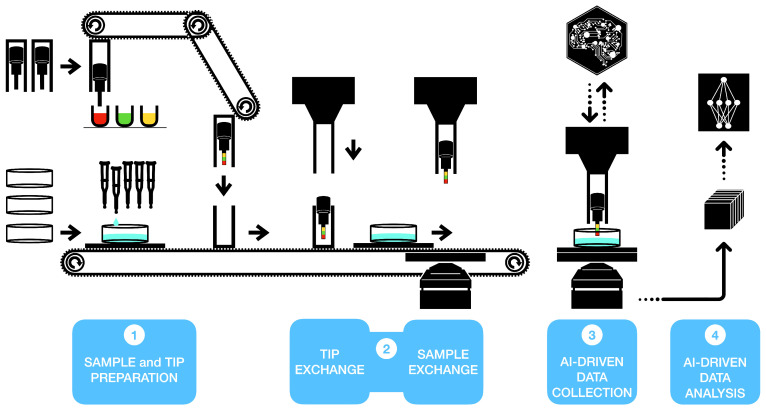
How the VPS and HPS could unlock the four automation levels described in the text. A conventional automated micro-pipetting combined with a conveyor system can directly prepare and deliver samples to the microscope (level 1 and 2). Similarly, the VPS can be used for probe functionalisation, while the same conveyor belt design can deliver the probe to the microscope head (see Appendix A). AI can control the full motorised microscope for data acquisition (3) and perform the data analysis (4).

## Data Availability

Data presented in this study is available on request from the corresponding author.

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
