# Peer review of "Towards a Fully Automated Scanning Probe Microscope for Biomedical Applications"

_sensors, 2021, doi:10.3390/s21093027_

Round 1

Reviewer 1 Report

A more detailed description of the actions of the automation is required in Figure 7.

More detail about the proposed use of this system in a biological application is needed in section 3.4. It is difficult to determine from the limited information whether the proposed setup will address the issues associated with moving and manipulating cells in an automated system.

Reviewer 2 Report

This is a very interesting work reporting on the development of a novel SPM design with sub-femtoNewton force sensitivity and low mechanical drift. The paper is clearly written, results and arguments are well supported and the paper can be accepted for publication in Micromachines. I only would like to ask authors to include an actual picture of the SPM setup they built alongside Fig.3.

Reviewer 3 Report

The manuscript presented by Szeremeta et al. describes the results on the bottom-up development of the vertically oriented probe modification of the SPM method. The combination of positioning stages, orientation and detection of the probe produces an SPM design compatible with the full automation is discussed. The proposed probe microscope sensitivity achieves sub-femtoNewton force whilst preserving low mechanical drift (2.0 ± 0.2nm/min in-plane and 1.0 ± 0.1nm/min vertically).

The article adheres to the journal’s standards, has an appropriate writing style, and figures appearance.

Besides, some issues have to be clarified.

  • Is the proposed design implemented into the operating microscope, or there is a prototype? It should be mentioned directly in the body of the manuscript. If implemented, the example of the scan images acquired in air and liquid environments is demanded.
  • The VPS design does not clearly describe how the piezoelectric stack actuator moves the stainless steel cylindrical plug into the glass tube. May authors describe the mechanism in details.
  • The principle of the feedback system of the proposed modification of SPM during the scanning procedure is not evident. May author add the missed information.
  • Does the cantilever oscillation approach shown in Fig. 4 have equal Q-factor and sensitivity in air and liquid? The information is missed regarding this comparison.
  • Figure 5a shows the smallest steps obtained in the air. Do the smallest steps in liquid conditions have the same values?
  • Figure 6 shows the shift in the distribution of positions of a thermally driven cantilever when illuminating with the laser. This gives rise to doubt due to the question of simultaneous thermal expansion of the media around the probe caused by the same laser heating.
  • The potential use with extreme force resolution in a liquid environment is very demanded in the biological application, but it is described very vaguely in the manuscript. Can authors provide more details regarding the measurements in a liquid environment?

Reviewer 4 Report

The authors attempted to a complete automation of lateral force scanning probe microscopy. From the preparation of the probes to data analysis using AI. If this system is realized, the application of LFSPM will spread into various biomedical applications. The reviewer recommends the acceptance of this article after minor corrections.

  1. The reviewer thinks that LMFM is not so popular compared with conventional AFM or optical tweezers. Is it possible to state the limitation of this method compared with the conventional ones? If the authors explain guidelines for the chemical design of probe and substrate (fixation of samples on the substrate, choice of linkers, etc.). The readers may be interested in the measurements of ligand-receptor systems with this method, whose force-sensitivity is very high.
  2. The authors mentioned AI-based automation of the measurements by citing several papers. However, references described the automation of AFM with the conventional configurations. The author should explain more details of the automation of the operation and data analysis.

Round 2

Reviewer 3 Report

The revised version of the manuscript can be accepted.